# Independent Negative Prognostic Role of TCF1 Expression within the Wnt/β-Catenin Signaling Pathway in Primary Breast Cancer Patients

**DOI:** 10.3390/cancers11071035

**Published:** 2019-07-22

**Authors:** Concetta Saponaro, Emanuela Scarpi, Francesco Alfredo Zito, Francesco Giotta, Nicola Silvestris, Anita Mangia

**Affiliations:** 1Functional Biomorphology Laboratory, IRCCS Istituto Tumori “Giovanni Paolo II” of Bari, 70124 Bari, Italy; 2Unit of Biostatistics and Clinical Trials, (IRST)-IRCCS Istituto Scientifico Romagnolo per lo Studio e la Cura dei Tumori, 47014 Meldola (FC), Italy; 3Pathology Department, IRCCS Istituto Tumori “Giovanni Paolo II” of Bari, 70124 Bari, Italy; 4Medical Oncology Unit, IRCCS-Istituto Tumori “Giovanni Paolo II” of Bari, 70124 Bari, Italy; 5Department of Biomedical Sciences and Human Oncology, University of Bari ‘Aldo Moro’, 70124 Bari, Italy

**Keywords:** Breast Cancer, Wnt pathway, TCF1, NHERF1

## Abstract

The Wnt pathway is involved in the progression of breast cancer (BC). We aimed to evaluate the expression of some components of the Wnt pathway (β-catenin, FZD4 (frizzled receptor 4), LRP5 (low-density lipoprotein receptor-related protein 5), LRP6, and TCF1 (T-cell factor 1)) to detect potential associations with NHERF1 (Na+/H+ exchanger regulatory factor 1) protein. Besides, we assessed their impact on patients’ clinical outcome. We evaluated 220 primary BC samples by immunohistochemistry (IHC) and protein localization by immunofluorescence. We found a significant correlation between NHERF1 and FZD4, LRP5, LRP6, and TCF1. Univariate analysis showed that the overexpression of β-catenin (*p* < 0.0001), FZD4 (*p* = 0.0001), LRP5, LRP6, and TCF1 (*p* < 0.0001 respectively) was related to poor disease-free survival (DFS). A Kaplan-Meier analysis confirmed univariate data and showed a poor DFS for cNHERF1+/FZD4+ (*p* = 0.0007), cNHERF1+/LRP5+ (*p* = 0.0002), cNHERF1+/LRP6+ (*p* < 0.0001), and cNHERF1+/TCF1+ phenotypes (*p* = 0.0034). In multivariate analysis, the expression of TCF1 and β-catenin was an independent prognostic variable of worse DFS (*p* = 0.009 and *p* = 0.027, respectively). In conclusion, we found that the overexpression of β-catenin, FZD4, LRP5, LRP6, and TCF1 was associated with poor prognosis. Furthermore, we first identified TCF1 as an independent prognostic factor of poor outcome, indicating it as a new potential biomarker for the management of BC patients. Also, the expression of Wnt pathway proteins, both alone and in association with NHERF1, suggests original associations of biological significance for new studies.

## 1. Introduction

Breast Cancer (BC) is one of the most common malignant cancers in women [1]; this makes it important to better understand the molecular mechanisms that underly the evolution and the possible interaction between the different molecules, which play a role in its development and progression. 

Na+/H+ exchanger regulatory factor 1 (NHERF1) is a scaffold protein, formed by two tandem PDZ (post-synaptic density 95/discs large/zona occludens 1) domains and a carboxyl-terminal ezrin-binding (EB) region [2]. This structure makes NHERF1 a sticky protein, able to interact with various signal molecules and implicated in different diseases [3]. In the last decade, a lot of studies highlighted its involvement during carcinogenesis and tumor progression [4,5,6,7,8,9], in particular, in BC [10,11,12]. A previous study, which we carried out demonstrated the prognostic significance of nuclear NHERF1 expression (nNHERF1) in a cohort of patients with primary BC [13]. It is worth mentioning that NHERF1 plays a significant role during the regulation of several oncogenic signaling pathways, including Wnt/β-catenin [14,15]. Wnt/β-catenin pathway is an evolutionarily conserved pathway that controls cell proliferation, differentiation, and the maintenance of stem cells [16]. The WNT/β-catenin pathway is modulated from transcriptional to post-transcriptional modifications. Its aberrant signaling is observed in cancers, stimulating the expression of numerous target genes involved in tumor development [17]. The β-catenin activity is deregulated in numerous cancer diseases, and it is a promising therapeutical target [18]. Its role is already well-established in colorectal cancer, even though no effective therapies have been identified. Recently, the NHERF1-Wnt axis has been considered as a possible interactive network in different types of cancer. An in vivo model showed a possible role of NHERF1 for intestinal neoplasia through a suppressor activity upstream of Wnt-β-catenin [19]. The suppressor function of Wnt-β-catenin on NHERF1 by TCF4 interaction and promoter activity by T-cell factor 1 (TCF1)-*Nherf1* promoter gene interaction has been reported in an in vitro model of colorectal cancer [4]. No similar studies have been conducted on a BC model.

Wnt/β-catenin signaling activation needs both Frizzled receptors (FZDs) and low-density lipoprotein receptor-related protein (LRP) 5 and 6 co-receptors. Ten human isoforms of FZD receptors are known with seven transmembrane domains, and the majority terminates with a PDZ binding region [20]. The LRP5/6 also belong to a family of 10 LRP isoforms, with a pivotal role in endocytosis, cellular communication, embryonic development, lipid homeostasis, and disease [21]. The molecular interaction between NHERF1 and FZD receptors and their modulation activity have been studied [14], but little is known about NHERF1-LRP co-receptors interaction in physiological and pathological conditions. 

One of the downstream transcription factors of Wnt signaling is TCF1, a T-cell factor (TCF) [16]. Recent evidence showed an interaction between TCF1 and nherf1 promoter, interfering with WNT/β-catenin signaling [4].

In this research, we analyzed the expression and the relationship among NHERF1 and some of the major important players of the WNT pathway, such as β-catenin, FZD4, LRP5, LRP6, and TCF1 in a cohort of primary BCs. In this study regarding biomarkers, our primary objective was to correlate, for the first time, NHERF1 with the expression of β-catenin, FZD4, LRP5, LRP6, and TCF1 and to verify if an interaction exists between NHERF1 and these proteins in BC samples. A secondary goal was to verify their prognostic potential and their impact on patients’ clinical outcome. 

A full understanding of the action and interaction of these biomarkers could improve current prognostic and therapeutic approaches and contribute to finding new ways, ensuring more precise BC management. 

## 2. Results

### 2.1. Profiling of Expression and Relationship between Tumor Markers and Clinicopathological Features

Forty-eight percent (106/220) of tumors presented higher NHERF1 expression in the cytoplasmic compartment (cNHERF1) as compared to the 10% present in nuclear staining (nNHERF1) (23/220). Cytoplasmic β-catenin expression was evaluated in the whole cohort with high cytoplasmic staining in the 50% of samples (110/220). FZD4 was overexpressed in 50.4% of patients (111/220). We found a predominant cytoplasmic expression, both of LRP5 and LRP6, which resulted in the overexpression in 48.6% (107/220) and 44.5% (98/220) of tumor samples, respectively. Finally, nuclear TCF1 overexpression was present in 51% of the cases (112/220). Figure 1 (panel A) shows an example of the immunohistochemical staining pattern of the analyzed proteins. The relationship between the expression of NHERF1, β-catenin, FZD4, LRP5, LRP6, and TCF1 and the clinicopathological characteristics are listed in Table 1A,B. We found an increase in cNHERF1 expression in tumors >2 cm (*p* = 0.0020), and a low nNHERF1 expression in tumors with a higher histological grade (G3), (*p* = 0.0386). Cytoplasmic β-catenin expression was observed in Invasive Ductal Carcinoma (IDCs) (*p* = 0.0138), and it was related to estrogen receptor (ER)-negative (*p* < 0.0001), progesterone receptor (PR)-negative (*p* = 0.0050), human epidermal growth factor receptor 2 (HER2)/neu-negative (*p* = 0.0002), and high proliferative activity (Ki67 index) (*p* = 0.0082). FZD4 was overexpressed in IDCs (*p* = 0.0199), in tumors >2 cm (*p* = 0.0010), and in tumors with a high histological grade (*p* = 0.0004). LRP5 was also overexpressed in tumors >2 cm (*p* = 0.0009), in high histological grade (*p* = 0.0002), and high Ki67 index (*p* = 0.0049). LRP6 was overexpressed in older patients (*p* = 0.0175), in tumors >2 cm (*p* = 0.0089), in G3 tumors (*p* = 0.0010), and high Ki67 index (*p* = 0.0235). LRP6 was inversely related to ER (*p* = 0.0012) and PR status (*p* = 0.0043). TCF1 expression was directly related to ER expression (*p* = 0.0432) (Table 1B).

### 2.2. Analysis of Association between NHERF1 and β-catenin, FZD4, LRP5, LRP6, TCF1 Protein Expression

We evaluated the relationship between cNHERF1 and nNHERF1 and the other biomarkers on continuous and dichotomized variables. The Spearman correlation test on continuous variables revealed a direct relation between cNHERF1 and FZD4 (rs: 0.34; *p* < 0.0001), LRP5 (rs: 0.24; *p* = 0.0022), and LRP6 (rs: 0.26; *p* = 0.0010). Besides, a direct correlation between cNHERF1and TCF1 was also observed (rs: 0.18; *p* = 0.0169). Further, nNHERF1 was inversely related to LRP5 (rs: –0.21; *p* = 0.0079), LRP6 (rs: −0.25; *p* = 0.0017), and TCF1 (rs: –0.21; *p* = 0.0048) (Table 2). No significant association between NHERF1 and β-catenin was found. These data were confirmed for dichotomized variables using the χ2 test (data not shown).

### 2.3. Immunofluorescence Co-Localization

We identified protein localization by immunofluorescence assay. Firstly, we detected the intracellular distribution of NHERF1 and β-catenin in an invasive cellular cluster of BCs. No co-localization was observed for these proteins (Figure 1, panel b,a).

Cytoplasmic NHERF1 and FZD4 proteins co-localized when they were found overexpressed within the cytoplasmic compartment (Figure 1, panel B-b). Immunofluorescence also provided the co-localization of cNHERF1 both with LRP5 and LRP6 (Figure 1, panel b,c,d, respectively). No co-localization was observed for NHERF1 and TCF1 (Figure 1, panel b–e). 

### 2.4. Expression of Proteins and Patient Clinical Outcome

Univariate and multivariate analyses were carried out including the expression of cNHERF1, nNHERF1, β-catenin, FZD4, LRP5, LRP6, and TCF1 proteins and all clinicopathological characteristics, as dichotomized variables.

The subgroup of patients with low β-catenin expression had a better 5-year % disease-free survival (DFS) compared to patients with high β-catenin expression 92% vs. 67% (*p* < 0.0001). Patients with high FZD4 expression had a worse 5-years % DFS compared with patients with low FZD4 expression (71% vs. 91% *p* = 0.0001). A worse 5-years % DFS for patients with high compared to low LRP5 expression (70% vs. 93%; *p* < 0.0001) and for patients with high compared to low LRP6 expression (68% vs. 95%; *p* < 0.0001) was also observed. Furthermore, a worse 5-years % DFS was found for patients with high compared to low TCF1 expression (68% vs. 92%; *p* < 0.0001). No significant results emerged from cNHERF1, nNHERF1, and overall survival (OS) analyses. In addition, univariate analysis indicated a worse 5-years % DFS for tumor size >2 cm respect to ≤2 cm (68% vs. 88%; *p* = 0.002), positive compared to negative nodal status (72% vs. 87%; *p* = 0.001), high (G3) compared to low (G1–2) histological grade (63% vs. 94%; *p* < 0.0001), negative compared to positive ER (58% vs. 92%; *p* < 0.0001) and PR (68% vs. 90%; *p* < 0.0001) expression, high compared to low Ki67 (68% vs. 90; *p* < 0.0001). Then, a poor OS was observed for high (G3) compared to low (G1-2) histological grade (90% vs. 99%; *p* = 0.026), negative compared to positive ER expression (88% vs. 99%; *p*=0.024), and high compared to low Ki67 (90% vs. 99; *p* = 0.004) (Table 3).

Multivariate analysis, according to the Cox proportional hazard regression model, showed that the expressions of β-catenin and TCF1 were independent prognostic factors, associated to a shorter DFS (Hazard Ratio (HR) = 3.26, 95% Confidence Interval (CI) 1.14–9.33, *p* = 0.027; HR = 4.86, 95% CI 1.47–16.05, *p* = 0.009, respectively). Further, high grade (G3) was also an independent prognostic marker for DFS (HR = 5.28, 95% CI 1.60–17.4, *p* = 0.006) (Table 4). 

Kaplan-Meier analysis confirmed the univariate data, and it didn’t show significant differences in the DFS between patients with high and low cNHERF1 protein expression (*p* = 0.9427) (Figure 2a); a poor DFS in the subgroup of patients with high β-catenin expression, high FZD4 expression, high LRP5 and LRP6 expression and in the subgroup of patients with high TCF1 expression (*p* < 0.0001 for each group) (Figure 2b–f) was observed.

Moreover, a poor DFS was observed for the following phenotypes: cNHERF1−/FZD4+ and cNHERF1+/FZD4+ (*p* = 0.0007), cNHERF1−/LRP5+ and cNHERF1+/LRP5+ (*p* = 0.0002), cNHERF1−/LRP6+ and cNHERF1+/LRP6+ (*p* < 0.0001), and cNHERF1−/TCF1+ and cNHERF1+/TCF1+ (*p* = 0.0034) (Figure 3a–d).

## 3. Discussion

Understanding the expression of proteins involved in cancer development could improve knowledge of the pathways that contribute to BC onset and progression. Among the various signaling pathways related to cancer, an important role is played by the Wnt/β-catenin pathway. This pathway is a central actor of embryo development and maintenance of cellular homeostasis. Its deregulation has been associated with different human diseases, including cancer [22]. In BC, β-catenin expression in the cytoplasm/nucleus has proven to be a significant prognostic factor [23], and the interaction between β-catenin and NHERF1 has been reported by different authors [14,24]. 

In this study, we analyzed the expression of some of the major constituents of the Wnt pathway (β-catenin, FZD4, LRP5, LRP6, TCF1) and their relationships with the scaffold protein NHERF1. Furthermore, we evaluated the expression of these markers and the relation with clinical-pathological characteristics and their impact on the patients’ survival. 

NHERF1 is an important regulator of different proteins, trafficking and localized as a cystic fibrosis (CF) transmembrane conductance regulator, ezrin and β-catenin. It has an oncogenic role when it is localized in the plasmatic membrane, and it acts as an oncosuppressor when it moves into the cytoplasmic/nuclear compartment [25,26].

β-catenin has pivotal roles in morphogenesis and human cancer because it has a double function both in adhesive complexes and as a transducer/transcriptional regulator in several signal transduction pathways. β-catenin mutations have not been found in BC despite its upregulation and association with other markers and with the poor clinical outcome [27]. β-catenin signaling is initiated through the formation of an FZD/LRP5/6 receptor complex, and the activation of this protein complex seems to have a crucial role in BC development and progression.

Frizzled proteins are found at the level of the plasmatic membrane. They are sited at the surface of Wnt-responsive cells, but they have been observed co-opted in the cytoplasmic compartment as part of a regulatory mechanism of the extracellular level of Wnt protein [28].

Little is known about the cancer tissue expression of FZD4, LRP5, LRP6 proteins, and we assessed their behavior for the first time in a clinical BC cohort. Interestingly, these proteins showed a higher expression in our series of BC samples mislocalized in the cytoplasm. Both LRP5 and LRP6 transcriptomic analysis showed high RNA levels in a subgroup of TNBCs (Triple negative breast cancers) [29]. Moreover, LRP6 overexpression had been associated to a more aggressive BC phenotype in a smaller cohort of patients, and its down-regulation was sufficient to inhibit tumorigenesis, suggesting it as a possible, new, promising therapeutic target [30]. 

The relation between these markers and some of the more aggressive clinicopathological characteristics (tumor size, histological grade, Ki67 index) highlighted their involvement in BC progression, by contributing in defining the role in cancer growth. 

The association analyses between NHERF1 and the other markers revealed a positive direct relation between cNHERF1 and FZD4, LRP5, LRP6, and TCF1 expression, but not with β-catenin, suggesting a possible novel tumor progression mechanism, that involves Wnt signaling components. Previous studies have reported a high cNHERF1 expression linked to tumor aggressive features, such as metastases, poor grade, and lymphovascular invasion [7,9]. The inverse relation among nNHERF1 and LRP5, LRP6, and TCF1 supported the results of our previous study [13]. Furthermore, we also observed, by immunofluorescence, the co-localization among the NHERF1 and FZD4, LRP5, LRP6 expression in the cytoplasmic compartment. NHERF1-FZD4 direct interaction has been previously reported by co-immunoprecipitation in an in vitro model of ovary cells [14]. We observed evidence of a similar distribution of NHERF1 and LRP5/LRP6 in the cytoplasm. The lack of co-localization of TCF1 and NHERF1 is not surprising because they act in different cellular compartments during the tumor progression [13,31,32]. Furthermore, high TCF1 expression was identified in over half of the cases, and it was related to ER expression, probably implicated in the modulation of potential target genes, as previously reported by El- Tanani and colleagues. They demonstrated that some of the different family members of the two sets of downstream transcription factors ERα/ERβ and Tcf-1/Tcf-4 interacted directly, modulating the promoter activity of target genes [33]. TCF1 has been also reported associated with the Nherf1 promoter in an “in vitro” model of colorectal cancer by β-catenin knockdown, increasing Nherf1 mRNA levels [4].

Multivariate analysis revealed the prognostic power of TCF1 and β-catenin, indicating them as independent biomarkers of prognosis for DFS in our cohort. Other authors reported a β-catenin expression associated with poor prognosis in invasive ductal carcinoma [34], and with unfavorable outcomes for BC patients [35]. We found a prognostic role for the transcription factor TCF1, and this probably could be related to its close relationship with β-catenin, with which it contributes to the recruitment of factors that create transcription “hot-spots” [36]. To our knowledge, this is the first study in which TCF1 constitutes a new and independent prognostic factor in BC, and it could be used to reveal new research possibilities in the panorama of specific pharmacological inhibition as part of targeted therapy.

The univariate analysis identified TCF1 and also β-catenin, FZD4, LRP5, and LRP6 poor prognosis biomarkers, associating them with a shorter 5-years % DFS. 

When we analyzed the outcome of patients with co-expression of cNHERF1 and other biomarkers, the cNHERF1+/FZD4+, cNHERF1+/LRP5+, cNHERF1+/LRP6+, and cNHERF1+/TCF1+ phenotypes showed a worse DFS. These results confirmed the discriminatory capability of FZD4, LRP5, LRP6, and TCF1 as prognostic markers and highlighted the weak discriminatory power of cNHERF1. It was not possible to identify the actual worst-case scenario, supporting the evidence of the previous behavior observed by our group [12,37]. TCF1 has proven itself once again a prognostic marker for DFS in BC also when co-expressed with cNHERF1. Our results were mainly about DFS, and not OS (Overall Survival), due to the low number of deaths in our cohort. 

## 4. Materials and Methods 

### 4.1. Patients and Clinicopathological Characteristics

This study was carried out on a retrospective series of 220 primary BCs, diagnosed between 1994 and 2012 at the IRCCS Institute, Istituto Tumori “Giovanni Paolo II” of Bari, Italy. The patients were selected retrospectively, according to the availability of the biological material and the clinical follow-up. Our patient series was not consecutive. All patients provided an informed consent form to use their biological tissue for research purposes, according to ethical standards. Patients were eligible if they had a histological diagnosis of invasive breast carcinomas of any size and no evidence of metastatic disease. Patients were excluded if they had a previous history of invasive breast cancer, or other previous or concomitant malignancies or concomitant diseases. The study was approved by the Ethics Committee of the Istituto Tumori “Giovanni Paolo II” with the reference 657/CE on 13th December 2018.

Supporting Information Appendix A summarizes the clinicopathological characteristics of the whole cohort. The median age of patients was 54 years (range 28–80). The median follow-up was of 64 months (range 6–235). Forty-nine patients (22.3%) developed a relapse. The tumor, node, metastasis (TNM) classification, tumor size, histological grade, estrogen receptor (ER), progesterone receptor (PR), proliferative activity (Ki67), and human epidermal growth factor receptor 2 (HER2) status were provided by the Pathology Department of our Institute. The classification of ER, PR, Ki67, and HER2 has been previously reported. HER2 status was classified as negative (score 0,1+, and 2+ not amplified) or positive (when scored 3+ by IHC or HER2 amplified by Fluorescence in situ hybridization (FISH), according to the guideline for BC ASCO/CAP-2007 [38]. 

### 4.2. Tissue Microarray and Immunohistochemistry.

Immunohistochemistry (IHC) was performed on tissue microarrays (TMAs) sections of 220 BC patients. TMAs were constructed using the Galileo Tissue MicroArrayer CK 4500 (Transgenomic, Omaha, USA) [12]. The TMA slides were processed and stained by the indirect immunoperoxidase method, using the BenchMark XT automated staining instrument (Ventana Medical Systems, Tucson, AZ, USA), as previously reported [12] or with a standard manual procedure [11]. 

Slides were probed with 1:350 anti-EBP50 rabbit polyclonal antibody (ThermoFisher Scientific, Rockford, IL, USA), 1:100 rabbit monoclonal antibody anti-β-catenin (clone: E247, Abcam, Cambridge, UK), 1:30 anti-Frizzled4 rabbit polyclonal antibody, 1:100 anti-LRP5 goat polyclonal antibody, 1:100 anti-LRP6 rabbit polyclonal antibody (all Abcam, Cambridge, UK), 1:50 anti-TCF1 rabbit monoclonal (Cell Signaling Technology, Danvers, MA, USA). The dilution of the primary antibodies was based on preliminary dilution experiments.

For automated staining method, the UltraView DAB IHC Detection Kit (Ventana Medical Systems, Tucson, AZ, USA) was used to detect NHERF1 and β-catenin protein expression. Tissues were counterstained with Hematoxylin II and Bluing Reagent for 12 min and 4 min, respectively. Samples were dehydrated by sequential washes, cleared in xylene, and then mounted.

For manual staining method, a polymer-based-IHC detection system was used as the amplification system (EnVision + System-HRP Labeled Polymer Anti-Rabbit or Anti-Mouse secondary antibody, Dako, Carpinteria, CA, USA), according to the manufacturer’s instruction. For LRP5 antibody, a donkey anti-goat horseradish peroxidase (HRP) conjugated was applied. The signaling was revealed by incubating the sections in 3,3′-diaminobenzidine (Liquid DAB + Substrate Chromogen System, Dako, Carpinteria, CA, USA) for 8–10 min. Cell nuclei were counterstained with Mayer’s Hematoxylin (Bio-Optica, Milano, Italy). Known positive controls were included in each staining run. All antibodies used in this study have been validated, and the procedures standardized in a pre-analytic phase. The omission of the primary antibody was used as negative controls. 

### 4.3. Immunohistochemical Assessment

Immunoreactivity was assessed independently by two observers, who were blinded to clinicopathological data. The results given by the two observers were identical in most cases, and discrepancies were resolved by re-examination and consensus. For all markers, NHERF1, β-catenin, FZD4, LRP5, LRP6, and TCF1, a three-field average percentage was assessed, and the median values of protein expression were considered as a cut-off. 

The immunostaining of NHERF1 and β-catenin was assessed, as previously described [12,15]. NHERF1 immunostaining was predominantly cytoplasmic, and, in some cases, intense nuclear staining was also observed. This was evaluated and scored separately. The cases were classified as positive, when cNHERF1 immunoreactivity was present in ≥80% (median value) of tumor cells and when nNHERF1 immunoreactivity was present in >0% (median value) of tumor cells.

Only cytoplasmic β-catenin immunostaining was considered, and the cases were classified as positive, when β-catenin immunoreactivity was present in ≥5% (median value) of tumor cells and nuclear staining was completely absent. FZD4 immunostaining was largely cytoplasmic. FZD4 was assessed by counting the number of immunoreactive cancer cells over total cancer cells (%) for fields (more of 500 tumor cells) at x400 magnification [39]. The immunoreactivity of LRP5 and LRP6 was observable as a brown cytoplasmic coloration, and a percentage of positive cells per field was reported. TCF1 was positive when strong nuclear staining was present (percentage of positive nuclei for field). For FZD4, LRP5, LRP6, and TCF1 proteins, the cases were classified positive when the immunoreactivity was present in ≥77%, ≥83%, ≥74%, and ≥16% (median values) of tumor cells, respectively.

### 4.4. Immunofluorescence

The immunofluorescence method was described previously [8]. In brief, formalin-fixed and paraffin-embedded tissue serial sections of 3 μm were deparaffinized and rehydrated in an ethanol series. Saline citrated buffer (pH 6.0) at 0.01 M at 95 °C for 30 min was used for antigen retrieval, then 0.1% Triton X100-Phosphate Buffered Saline was applied for 15 min. The blocking was performed with 1% Bovine Serum Albumin-Phosphate Buffered Saline for 30 min, and then the slides were incubated overnight at 4 °C with all the primary antibodies used for the immunohistochemistry assay together with a mouse anti-EBP50 (BD Transduction Laboratories, San Jose, CA; dilution 1:150). The Alexa Fluor 488 and Alexa Fluor 568 immunoglobulin G secondary conjugated antibodies (1:2000 dilution; Molecular Probes Inc., Eugene, OR, USA) were incubated at room temperature for 1 h, and then the slides were mounted with DAPI (ProLong® Gold antifade reagent; Molecular Probes Inc./ThermoFisher, Rockford, IL, USA). Positive control slides were run simultaneously to assess the quality of immunoreactivity. For negative controls, slide sections were treated with 1% Bovine Serum Albumin instead of the primary antibody, and no reactivity was observed in any of these controls. Images were obtained on an Axion Image 2 upright microscope (Zeiss, Oberkochen Germany) with an Axiocam 512 color camera. 

### 4.5. Follow-Up and Statistical Analysis 

The Chi-squared test was used for the analysis of the association between marker expression and clinicopathological characteristics. Spearman correlation from ranks was used to analyze the interaction between two continuous variables. These statistical evaluations were performed with the Prism version 5 software package (GraphPad Software, San Diego, CA, USA), with the statistical significance set at *p* < 0.05.

The analysis was carried out about disease-free survival (DFS) and overall survival (OS). DFS (in months) was calculated as the time-frame between the date of surgery and the date of loco-regional/distant relapse (second invasive BC, second primary cancer, and/or death without evidence of BC) to the date of the last contact. OS (in months) was calculated as the time-frame between the date of surgery and date of last contact or the date of death from any cause. DFS and OS survival curves were computed using the Kaplan-Meier method and compared by using the log-rank test. The Cox proportional hazard regression model was performed to assess prognostic factors, including the variables that were statistically significant in univariate analysis. The model was optimized using a backward stepwise regression. The statistical significance level was *p*-Values < 0.05. Statistical analyses were made using the SAS statistical software, version 9.4 (SAS Institute Inc, Cary, NC, USA).

## 5. Conclusions

The heterogeneous nature of breast cancer renders the identification of new prognostic biomarkers increasingly necessary. In this study, we analyzed the expression of biomarkers involved in the Wnt pathway to identify patients at risk of poor prognosis. We found that TCF1 was an independent prognostic factor of poor outcome, and we showed that the overexpression of the Wnt pathway proteins was associated with worse disease-free survival. These results suggest the possible capability of these proteins to stratify breast cancer patients. 

## Figures and Tables

**Figure 1 cancers-11-01035-f001:**
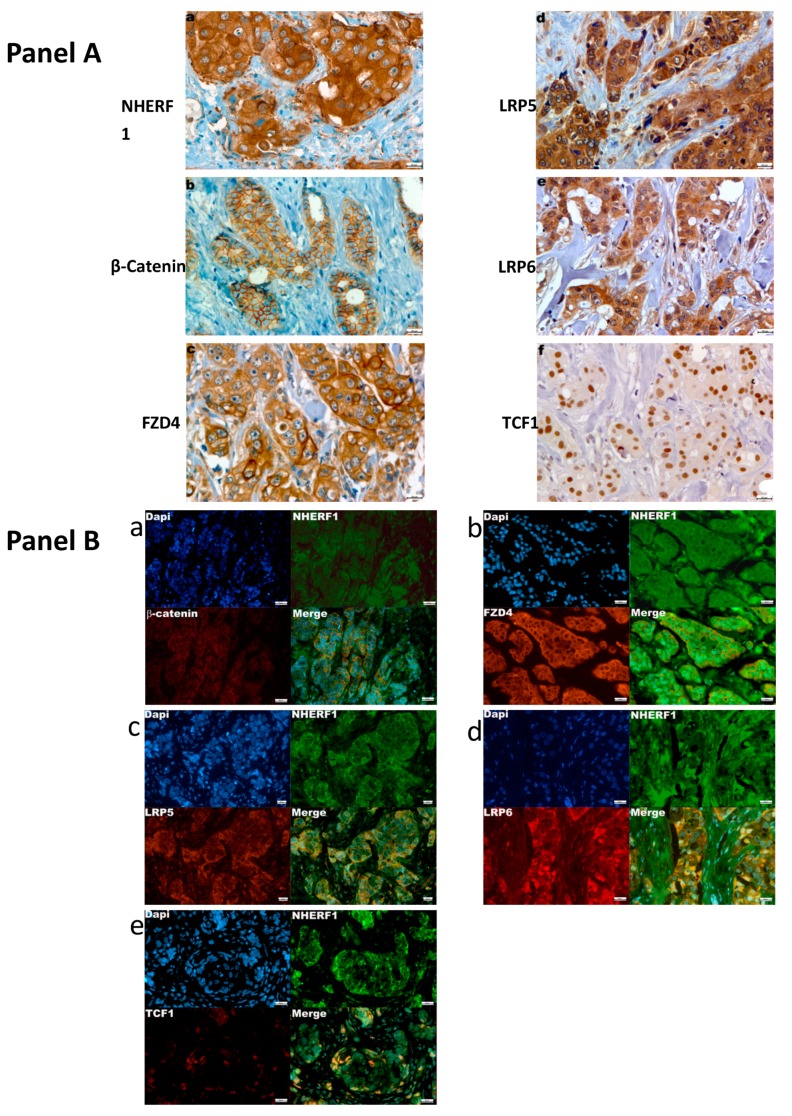
**Panel A.** Representative images of immunohistochemical staining in Breast Cancer tissues. The panel displays the representative expression of molecular biomarkers in tumor zone: (**a**) positive high cytoplasmic Na+/H+ Exchanger Regulatory Factor 1 (NHERF1) expression; (**b**) membranous and cytoplasmic β-catenin expression; (**c**) cytoplasmic Frizzled receptor 4 (FZD4) overexpression; (**d**) and (**e**) cytoplasmic Low-density lipoprotein receptor-related protein (LRP) 5 and Low-density lipoprotein receptor-related protein (LRP) 6 overexpression; (**f**) nuclear T-cell factor 1 (TCF1) expression (original magnification, ×400). Scale bar = 20 µm. **Panel B.** Immunofluorescence assay of the expression of NHERF1 and Wnt/pathway proteins. Representative tumor tissue samples stained with: (**a**) NHERF1 and β-catenin; (**b**) NHERF1 and FZD4; **c)** NHERF1 and LRP5; (**d**) NHERF1 and LRP6; (**e**) NHERF1 and TCF1 primary antibodies and detected with Alexa Fluor 488 (green) and Alexa Fluor 568 (red) secondary antibodies. The nuclei were stained with DAPI (4′,6-diamidino-2-phenylindole, blue) (original magnification, ×400). Scale bar= 20 µm. Images were obtained on an Axion Image 2 upright microscope (Zeiss, Oberkochen, Germany) with an Axiocam 512 color camera.

**Figure 2 cancers-11-01035-f002:**
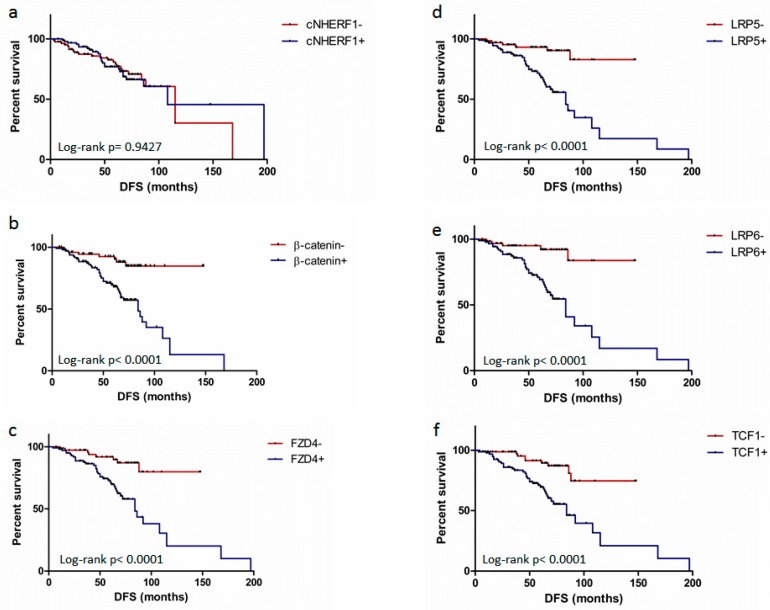
Survival analyses. Disease-free survival (DFS) curves for patients with (**a**) (cytoplasmic Na+/H+ Exchanger Regulatory Factor 1) cNHERF1– versus cNHERF1+ expression (*p* = 0.9427); (**b**) β-catenin- versus β-catenin+ expression (*p* < 0.0001); (**c**) (Frizzled receptor 4)FZD4– versus FZD4+ expression (*p* < 0.0001); (**d**) (Low-density lipoprotein receptor-related protein LRP 5) LRP5– versus LRP5+ (*p* < 0.0001); (**e**) (Low-density lipoprotein receptor-related protein LRP 6) LRP6– versus LRP6+ (*p* < 0.0001); (**f**) T-cell factor 1 (TCF1) TCF1– versus TCF1+ (*p* < 0.0001).

**Figure 3 cancers-11-01035-f003:**
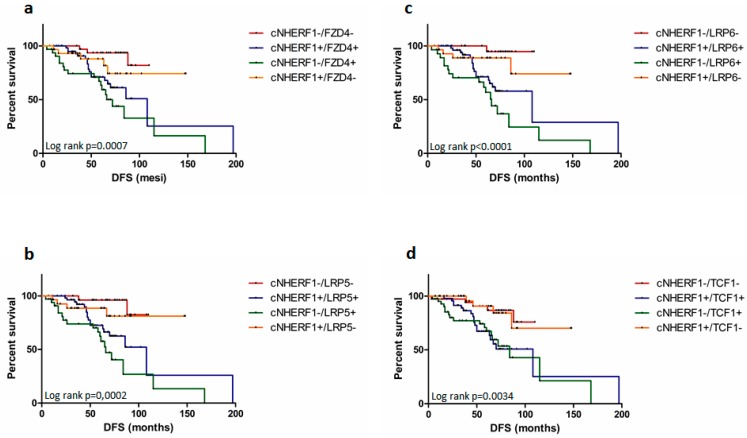
Survival analyses, according to both (Na+/H+ Exchanger Regulatory Factor 1) NHERF1 and Wnt/pathway proteins co-expression. Disease-free survival (DFS) curves for patients with (**a**) cNHERF1/FZD4 (Frizzled receptor 4) co-expression, four subgroups were compared: cNHERF1−/FZD4−, cNHERF1+/FZD4+, cNHERF1−/FZD4+, and cNHERF1+/FZD4− with a *p* = 0.0007; (**b**) cNHERF1/LRP5 (Low-density lipoprotein receptor-related protein LRP) co-expression, four subgroups were compared: cNHERF1−/LRP5-, cNHERF1+/LRP5+, cNHERF1−/LRP5+, and cNHERF1+/LRP5- with a *p* = 0.0002; (**c**) cNHERF1/LRP6 co-expression, four subgroups were compared: cNHERF1−/LRP6−, cNHERF1+/LRP6+, cNHERF1−/LRP6+, and cNHERF1+/LRP6− with a *p* < 0.0001; (**d**) cNHERF1/TCF1 (T-cell factor 1 (TCF1) co-expression, four subgroups were compared: cNHERF1−/TCF1−, cNHERF1+/TCF1+, cNHERF1−/TCF1+, and cNHERF1+/TCF1− with a *p* = 0.0034.

**Table cancers-11-01035-t001a:** (**A**)

	cNHERF1					nNHERF1					β-catenin					FZD 4				
	Negative		Positive		*p*-Value	Negative		Positive		*p-*Value	Negative		Positive		*p*-Value	Negative		Positive		*p-*Value
Characteristics	n	(%)	n	(%)		n	(%)	n	(%)		n	(%)	n	(%)		n	(%)	n	(%)	
**Patient age**																				
≤54years	49	(26.1)	50	(26.6)	0.0865	87	(46)	12	(6.4)	0.9831	39	(20.2)	58	(30)	0.4298	46	(23.8)	51	(26.4)	0.1632
>54 years	33	(17.5)	56	(29.8)		79	(41.8)	11	(5.8)		44	(22.8)	52	(27)		36	(18.7)	60	(31.1)	
**Histological type**																				
IDC	69	(36.7)	92	(48.9)	0.7512	141	(74.6)	21	(11.1)	0.2613	69	(35.8)	101	(52.3)	**0.0138**	65	(33.7)	103	(53.4)	**0.0199**
ILC	7	(3.7)	9	(4.9)		16	(8.5)	0	/		10	(5.2)	2	(1)		9	(4.7)	5	(2.6)	
Other	6	(3.2)	5	(2.6)		9	(4.8)	2	(1)		4	(2)	7	(3.7)		8	(4.1)	3	(1.5)	
**Tumor size**																				
≤2 cm	58	(32.2)	47	(26.2)	**0.0020**	94	(52)	12	(6.6)	0.5055	50	(26.8)	60	(32.2)	0.2420	57	(30.6)	53	(28.5)	**0.0010**
>2 cm	24	(13.3)	51	(28.3)		64	(35.4)	11	(6)		28	(15)	48	(26)		21	(11.3)	55	(29.6)	
**Lymph node status**																				
Negative	50	(26.9)	61	(32.8)	0.7486	98	(53.7)	13	(7)	0.7417	51	(27.3)	61	(32.6)	0.3521	49	(26.2)	61	(32.6)	0.5599
Positive	32	(17.2)	43	(23.1)		65	(35)	10	(5.3)		29	(15.5)	46	(24.6)		31	(16.6)	46	(24.6)	
**Histological grade**																				
G1–2	49	(26.8)	49	(26.8)	0.1294	82	(44.6)	17	(9.2)	**0.0386**	49	(26)	54	(28,7)	0.0896	56	(30)	47	(25)	**0.0004**
G3	33	(18)	52	(28.4)		79	(42.9)	6	(3.3)		30	(16)	55	(29,3)		24	(12.8)	60	(32.2)	
**Estrogen Receptor**																				
ER-negative (≤10%)	26	(14)	31	(16.7)	0.7802	51	(27.3)	7	(3.7)	0.9487	13	(6.9)	48	(25.4)	**<0.0001**	20	(10.6)	39	(20.6)	0.1141
ER-positive (>10%)	56	(30.1)	73	(39.2)		113	(60.4)	16	(8.6)		67	(35.4)	61	(32.3)		60	(31.7)	70	(37.1)	
**Progesterone Receptor**																				
PR-negative (≤10%)	37	(19.9)	49	(26.3)	0.7866	77	(41.2)	10	(5.3)	0.7545	29	(15)	62	(33)	**0.0050**	34	(18)	54	(28.6)	0.3376
PR-positive (>10%)	45	(24.2)	55	(29.6)		87	(46.5)	13	(7)		51	(27)	47	(25)		46	(24.3)	55	(29.1)	
**HER2/neu**																				
Negative (0,1+)	70	(38.2)	89	(48.6)	0.8281	143	(77.8)	17	(9.2)	0.7832	68	(39.3)	94	(54.3)	**0.0002**	66	(35.5)	96	(51.6)	0.3909
Positive (3+)	10	(5.5)	14	(7.7)		21	(11.4)	3	(1.6)		11	(6.4)	0	/		12	(6.4)	12	(6.4)	
**Ki67**																				
Negative (≤ 20%)	48	(26)	54	(29.1)	0.4065	91	(49)	12	(6.4)	0.7414	54	(28.7)	52	(27.7)	**0.0082**	52	(27.6)	55	(29.3)	0.0540
Positive (> 20%)	34	(18.4)	49	(26.5)		72	(38.7)	11	(5.9)		26	(13.8)	56	(29.8)		28	(14.9)	53	(28.2)	

**Table cancers-11-01035-t001b:** (**B**)

	LRP5					LRP6					TCF1				
	Negative		Positive		*p-*Value	Negative		Positive		*p-*Value	Negative		Positive		*p-*Value
Characteristics	n	(%)	n	(%)		n	(%)	n	(%)		n	(%)	n	(%)	
**Patient age**															
≤54 years	37	(20.6)	54	(30)	0.9771	45	(26.6)	44	(26)	**0.0175**	45	(23)	59	(30)	0.9014
>54 years	36	(20)	53	(29.4)		26	(15.4)	54	(32)		39	(20)	53	(27)	
**Histological type**															
IDC	61	(33.9)	96	(53.3)	0.4697	59	(35)	89	(52.6)	0.1696	74	(37.8)	94	(48)	0.1396
ILC	7	(3.9)	6	(3.3)		5	(3)	6	(3.5)		3	(1.5)	12	(6.1)	
Other	5	(2.8)	5	(2.8)		7	(4.1)	3	(1.8)		7	(3.6)	6	(3)	
**Tumor size**															
≤2 cm	54	(31)	51	(29.3)	**0.0009**	51	(31.3)	49	(30)	**0.0089**	50	(26.5)	64	(33.8)	0.8714
>2 cm	18	(10.3)	51	(29.3)		19	(11.7)	44	(27)		32	(17)	43	(22.7)	
**Lymph node status**															
Negative	42	(24.1)	63	(36.2)	0.6486	42	(25.6)	59	(36)	0.7187	50	(26.3)	60	(31.6)	0.5640
Positive	30	(17.2)	39	(22.5)		28	(17)	35	(21.4)		33	(17.4)	47	(24.7)	
**Histological grade**															
G1–2	50	(28.7)	42	(24.1)	**0.0002**	47	(28.7)	40	(24.4)	**0.0010**	49	(25.8)	56	(29.5)	0.2778
G3	22	(12.7)	60	(34.5)		22	(13.4)	55	(33.5)		33	(17.4)	52	(27.3)	
**Estrogen Receptor**															
ER-negative (≤10%)	19	(10.8)	38	(21.6)	0.1571	12	(7.2)	39	(23.5)	**0.0012**	20	(10.4)	42	(21.9)	0.0432
ER-positive (>10%)	53	(30.1)	66	(37.5)		58	(35)	57	(34.3)		62	(32.3)	68	(35.4)	
**Progesterone Receptor**															
PR-negative (≤10%)	27	(15.3)	53	(30.1)	0.0778	23	(13.8)	53	(31.9)	**0.0043**	34	(17.7)	57	(29.7)	0.1552
PR-positive (>10%)	45	(25.6)	51	(29)		47	(28.3)	43	(26)		48	(25)	53	(27.6)	
**HER2/neu**															
Negative (0,1+)	58	(33.3)	94	(54)	0.0619	60	(36.1)	84	(50.6)	0.7376	63	(33.3)	100	(53)	0.0670
Positive (3+)	13	(7.5)	9	(5.2)		10	(6)	12	(7.3)		15	(8)	11	(5.8)	
**Ki67**															
Negative (≤20%)	51	(29.1)	51	(29.1)	**0.0049**	47	(28.5)	47	(28.5)	**0.0235**	42	(22)	66	(34.5)	0.1979
Positive (>20%)	21	(12)	52	(29.8)		23	(14)	48	(29)		40	(21)	43	(22.5)	

Bold values indicate significance. cNHERF1: cytoplasmic Na+/H+ Exchanger Regulatory Factor 1; nNHERF1: nuclear Na+/H+ Exchanger Regulatory Factor 1; FZD4: Frizzled receptor 4; LRP5: Low-density lipoprotein receptor-related protein (LRP) 5; LRP 6: Low-density lipoprotein receptor-related protein (LRP) 6; TCF1: T-cell factor 1 (TCF1); IDC: Invasive Ductal Carcinoma; ILC: Invasive Lobular Carcinoma; ER: Estrogen receptor; PR: Progesterone receptor; HER2/neu: Human epidermal growth factor receptor 2.

**Table 2 cancers-11-01035-t002:** Spearman for rank-based correlations between protein expression in BC (breast cancer) patients on continuous variables.

	β-catenin		FZD4		LRP5		LRP6		TCF1	
	*r*	*p-*Value	*r*	*p-*Value	*r*	*p-*Value	*r*	*p-*Value	*r*	*p-*Value
**cNHERF1**	0.01	0.883	0.34	**<0.0001**	0.24	**0.0022**	0.26	**0.0010**	0.18	**0.0169**
**nNHERF1**	0.047	0.5394	−0.12	0.1110	−0.21	**0.0079**	−0.25	**0.0017**	−0.21	**0.0048**

Spearman correlation coefficient r (Rho) and *p*-Value. Bold values indicate significance. cNHERF1: cytoplasmic Na+/H+ Exchanger Regulatory Factor 1; nNHERF1: nuclear Na+/H+ Exchanger Regulatory Factor 1; FZD4: Frizzled receptor 4; LRP5: Low-density lipoprotein receptor-related protein (LRP) 5; LRP 6: Low-density lipoprotein receptor-related protein (LRP) 6; TCF1: T-cell factor 1 (TCF1).

**Table 3 cancers-11-01035-t003:** Univariate analysis of DFS (disease-free survival) and OS (overall survival).

		DFS					OS				
Characteristics	N. pts	N. events	5-yrs % DFS	*p*-Value	HR (95% CI)	*p*-Value	N. events	5-yrs % OS	*p*-Value	HR (95% CI)	*p*-Value
**Overall**	200	51	80	-	-	-	10	95	-	-	-
**β-catenin**											
<5	75	9	92		1.00		2	98		1.00	
≥5	101	41	67	**<0.0001**	4.70 (2.20–10.04)	**<0.0001**	8	92	0.139	3.03 (0.64–14.28)	0.161
**FZD4**											
<77	76	9	91		1.00		3	97		1.00	
≥77	101	39	71	**0.0001**	3.76 (1.81–7.82)	**0.0004**	7	93	0.399	1.77 (0.46–6.87)	0.406
**LRP5**											
<83	66	6	93		1.00		1	98		1.00	
≥83	99	39	70	**<0.0001**	5.44 (2.29–12.96)	**0.0001**	8	91	0.068	5.56 (0.70–44.51)	0.106
**LRP6**											
<74	65	5	95		1.00		2	98		1.00	
≥74	92	38	68	**<0.0001**	6.14 (2.39–15.75)	**0.0002**	7	91	0.290	2.28 (0.47–10.98)	0.304
**TCF1**											
<16	78	9	92		1.00		2	100		1.00	
≥16	98	40	68	**<0.0001**	4.15 (2.00–8.61)	**0.0001**	7	91	0.165	2.89 (0.60–13.96)	0.185
**cNHERF1**											
<80	85	23	80		1.00		7	92		1.00	
≥80	96	25	77	0.981	1.01 (0.57–1.79)	0.981	3	97	0.160	0.39 (0.10–1.52)	0.177
**nNHERF1**											
0	159	43	80		1.00		7	96		1.00	
>0	22	5	84	0.621	0.79 (0.31–2.01)	0.623	3	89	0.124	2.77 (0.71–10.74)	0.141
**Age (years)**											
≤54	104	33	78		1.00		6	95		1.00	
>54	96	18	81	0.339	0.75 (0.42–1.36)	0.342	4	95	0.707	0.78 (0.22–2.78)	0.708
**Histotype**											
Ductal	172	45	79		1.00	0.673	9	95		1.00	0.727
Lobular	18	4	84		0.59 (0.18–1.94)		0	100		0	
Other	10	2	87	0.666	0.85 (0.20–3.54)		1	83	0.446	2.32 (0.29–18.43)	
**Tumor size (cm)**											
≤2.0	117	19	88		1.00		4	97		1.00	
>2.0	75	29	68	**0.002**	2.44 (1.36–4.37)	**0.003**	6	92	0.165	2.38 (0.67–8.45)	0.179
**Node**											
Negative	121	21	87		1.00		5	96		1.00	
Positive	75	26	72	**0.001**	2.58 (1.42–4.67)	**0.002**	5	93	0.359	1.77 (0.51–6.12)	0.366
**Grade**											
1-2	108	7	94		1.00		2	99		1.00	
3	87	42	63	**<0.0001**	7.50 (3.35–16.78)	**<0.0001**	8	90	0.026	4.87 (1.03–22.93)	**0.045**
**ER (%)**											
≤10	63	36	58		1.00		7	88		1.00	
>10	135	13	92	**<0.0001**	0.19 (0.10–0.37)	**<0.0001**	3	99	0.024	0.24 (0.06–0.93)	**0.039**
**PR (%)**											
≤10	92	38	68		1.00		8	91		1.00	
>10	106	11	90	**<0.0001**	0.28 (0.14–0.55)	**0.0002**	2	99	0.055	0.25 (0.05–1.16)	**0.077**
**Ki67 (%)**											
≤20	113	12	90		1.00		1	99		1.00	
>20	84	36	68	**<0.0001**	3.68 (1.90–7.13)	**0.0001**	9	90	0.004	11.25 (1.42–88.83)	**0.022**
**HER2**											
Negative	173	48	78		1.00		10	94		1.00	
Positive	26	2	88	0.137	0.36 (0.09–1.48)	0.155	0	100	0.275	0	0.993

Bold values indicate significance. HR: Hazard-ratio; FZD4: Frizzled receptor 4; LRP5: Low-density lipoprotein receptor-related protein (LRP) 5; LRP 6: Low-density lipoprotein receptor-related protein (LRP) 6; TCF1: T-cell factor 1 (TCF1); cNHERF1: cytoplasmic Na+/H+ Exchanger Regulatory Factor 1; nNHERF1: nuclear Na+/H+ Exchanger Regulatory Factor 1; ER: Estrogen receptor; PR: Progesterone receptor; HER2/neu: Human epidermal growth factor receptor 2.

**Table 4 cancers-11-01035-t004:** Multivariate analysis of DFS (disease-free survival) and OS (overall survival).

	DFS		OS	
	HR (95% CI)	*p*-Value	HR (95% CI)	*p*-Value
Grade (3 vs. 1–2)	5.28 (1.60–17.41)	**0.006**	-	-
β-catenin	3.26 (1.14–9.33)	**0.027**	-	-
TCF1	4.86 (1.47–16.05)	**0.009**	-	-

Bold values indicate significance. HR: Hazard-ratio; TCF1: T-cell factor 1(TCF1).

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
