# Peer review of "Independent Negative Prognostic Role of TCF1 Expression within the Wnt/β-Catenin Signaling Pathway in Primary Breast Cancer Patients"

_cancers, 2019, doi:10.3390/cancers11071035_

Round 1

Reviewer 1 Report

The article of Saponaro et al. entitled 'independent negative prognostic expression within wnt/beta-catenin signaling pathway in primary breast cancer patients' is a manuscript focusing on the relationship among NHERF1 and WNT players in breast cancers. the article is well written and interesting, conclusions are consistent with the results. Some comments could improve the manuscript: major point: - in the multivariate analysis; the biomarkers seems to have high correlations each others? is there relevant to do this analysis? could you more detail the relevance of the analysis and why TCF1 (in a molecular point of view) is an independent predictor- more detail the wnt pathway in the introduction minor point: - more detail the wnt pathway in introduction - an interesting paper for the role of wnt in cancers for your article : vallée et al.Crosstalk Between Peroxisome Proliferator-Activated Receptor Gamma and the Canonical WNT/β-Catenin Pathway in Chronic Inflammation and Oxidative Stress During Carcinogenesis. Frontiers in Immnology. 2018

Author Response

All changes to our manuscript within the document have been marked in yellow.

Reviewer 1

Open Review

English language and style

( ) Extensive editing of English language and style required
( ) Moderate English changes required
(x) English language and style are fine/minor spell check required
( ) I don't feel qualified to judge about the English language and style

Yes

Can be   improved

Must be   improved

Not applicable

Does the introduction provide sufficient background and include all   relevant references?

(x)

( )

( )

( )

Is the research design appropriate?

(x)

( )

( )

( )

Are the methods adequately described?

(x)

( )

( )

( )

Are the results clearly presented?

(x)

( )

( )

( )

Are the conclusions supported by the results?

(x)

( )

( )

( )

Comments and Suggestions for Authors

The article of Saponaro et al. entitled 'independent negative prognostic expression within wnt/beta-catenin signaling pathway in primary breast cancer patients' is a manuscript focusing on the relationship among NHERF1 and WNT players in breast cancers. the article is well written and interesting, conclusions are consistent with the results. Some comments could improve the manuscript:

major point: - in the multivariate analysis; the biomarkers seems to have high correlations each others? is there relevant to do this analysis? could you more detail the relevance of the analysis and why TCF1 (in a molecular point of view) is an independent predictor- more detail the wnt pathway in the introduction

Major point:

- In the multivariate analysis, Grade, β-catenin and TCF1 remain as independent prognostic factors for DFS, indicating that these variables are not correlated with each other and thus each of them provide information which is  independent as respect to other variables. For this analysis there is no relevance.

- As regards the  molecular role of TCF1 as an  independent predictor,  we emphasize the fact  that this is the first time in which a study provides the evidence that TCF1 is a negative independent predictor of disease free survival. Certainly, its role linked to the transcription of cancer-related genes is fundamental. Probably, the microenvironment influences the TCF1 expression, which exerts context-dependent functions. Further molecular studies are needed to better understand the  mechanism of its action.

Minor point:

minor point: - more detail the wnt pathway in introduction - an interesting paper for the role of wnt in cancers for your article : vallée et al.Crosstalk Between Peroxisome Proliferator-Activated Receptor Gamma and the Canonical WNT/β-Catenin Pathway in Chronic Inflammation and Oxidative Stress During Carcinogenesis. Frontiers in Immnology. 2018

- We thank the reviewer for the suggestion. We added the following  sentence and the relative reference in the introduction as suggested :

 “ The WNT/β-catenin pathway is modulated from transcriptional to post-transcriptional modifications. Its aberrant signaling is observed in cancers, stimulating the expression of numerous target genes involved in tumor development (Valleè A. and Lecarpentier Y., Front Immunol. 2018).”

Reviewer 2 Report

Summary

In the present manuscript Saponaro and colleagues set out to investigate the whether there is a correlation between NHERF1 and several components of the canonical Wnt Signaling pathway as well as “verify if an interaction exist between NHERF1 and these proteins in BC samples. The premise is that understanding the relationship between these proteins could importantly contribute with important insight into BC progression as well as aid the development of novel therapeutic options for BC patients. The association between high levels of Wnt activity in breast cancer and poorer prognosis has been studied before, thus diminishing the novelty of the present manuscript. In fact, several studies establish b-catenin as a marker of poorer prognosis. Nonetheless, the authors demonstrate for the first time a direct association between TCF1 expression, and to a lesser extent, LRP5/6 and FZD4, with poorer DFS in BC patients. However, this reviewer has several concerns regarding the way data is presented, the overall quality of images and figures, the style of writing and, most importantly with the criteria for patient selection.

Comments

Introduction

On lines 44-46, the authors highlight the importance of NHERF1 in regulating “several oncogenic signalling pathways, including Wnt/b-catening signalling”, but fail to elucidate how it does so, and what hypothetic importance it may or may not have in the context of cancer in general and breast cancer in particular.  Since a significant part of the manuscript focuses on investigating the co-expression of this protein with components of Wnt signalling, this reviewer would like to have broader clarification of what is currently known regarding the NHERF1-Wnt axis.

TCF1 expression and Estrogen Receptor positivity

On lines 87-88 the authors state that TCF1 expression was only directly related to ER positive tumors. However, on supplementary Table 2, they report that twice as many ER-/TCF1+ than ER-/TCF1- patients.

Figure 1.

Panel A – Pictures have very low resolution. Figure letters are a), c) and e) are not very clearly visible. Scale bar missing.

Panel B - Pictures have, again, low resolution and do not look sharp enough, making it very hard to appreciate cellular morphology and subcellular localization of the IF-labelled proteins. The picture legends are also small, smudged and hard to read. Scale bars seem to be present at the right-hand lower corner of each picture, but they are impossible to read.

cNHERF1 and nNHERF1 are only defined in line 279 and 280, yet these abbreviations are extensively used throughout the manuscript with no prior definition.

Patient selection

There is no information whatsoever concerning the selection criteria, if there was any, regarding treatment regimens and therapeutic agents’ patients were submitted to.

Subtype-specific prognostic value of the different Wnt signalling markers

This reviewer would appreciate a BC subtype specific survival analysis for the expression of each of the Wnt markers.

Immunohistochemical assessment

This reviewer would like a clarification on the criteria for the selection of Pos vs Neg thresholds for the IHC of the markers described in lines 274-291.

English:
The manuscript is intelligible. However, it should be reviewed by a scientific-writing expert to mitigate several syntax and grammatical issues which make the readthrough harder. Some examples are:

i)                 the consistent lack of articles (line 17: “(The) Wnt pathway is (…)”);

ii)                the use of informal writing style in some instances (line 41: “able to interact with a lot of (…)”);

iii)              poor syntax (line 100: “Association between Protein Expressions Analyzed”);

(…)

Author Response

All changes to our manuscript within the document have been marked in yellow.

Reviewer 2

Open Review

English language and style

(x) Extensive editing of English language and style required
( ) Moderate English changes required
( ) English language and style are fine/minor spell check required
( ) I don't feel qualified to judge about the English language and style

Yes

Can be   improved

Must be   improved

Not   applicable

Does the introduction provide sufficient background and include all   relevant references?

( )

( )

(x)

( )

Is the research design appropriate?

( )

(x)

( )

( )

Are the methods adequately described?

( )

(x)

( )

( )

Are the results clearly presented?

( )

( )

(x)

( )

Are the conclusions supported by the results?

( )

( )

(x)

( )

Comments and Suggestions for Authors

Summary

In the present manuscript Saponaro and colleagues set out to investigate the whether there is a correlation between NHERF1 and several components of the canonical Wnt Signaling pathway as well as “verify if an interaction exist between NHERF1 and these proteins in BC samples. The premise is that understanding the relationship between these proteins could importantly contribute with important insight into BC progression as well as aid the development of novel therapeutic options for BC patients. The association between high levels of Wnt activity in breast cancer and poorer prognosis has been studied before, thus diminishing the novelty of the present manuscript. In fact, several studies establish b-catenin as a marker of poorer prognosis. Nonetheless, the authors demonstrate for the first time a direct association between TCF1 expression, and to a lesser extent, LRP5/6 and FZD4, with poorer DFS in BC patients. However, this reviewer has several concerns regarding the way data is presented, the overall quality of images and figures, the style of writing and, most importantly with the criteria for patient selection.

Comments

Introduction

On lines 44-46, the authors highlight the importance of NHERF1 in regulating “several oncogenic signalling pathways, including Wnt/b-catening signalling”, but fail to elucidate how it does so, and what hypothetic importance it may or may not have in the context of cancer in general and breast cancer in particular.  Since a significant part of the manuscript focuses on investigating the co-expression of this protein with components of Wnt signalling, this reviewer would like to have broader clarification of what is currently known regarding the NHERF1-Wnt axis.

-          We added the following sentences and the relative references in the introduction section :

“Recently, the NHERF1-Wnt axis has been considered as a possible interactive network in different types of  cancer.  An in vivo model showed a possible role of NHERF1 for intestinal neoplasia through a suppressor activity upstream of Wnt-β-catenin [Georgescu MM et al., Neoplasia 2016]. A Wnt-β-catenin suppressor function on NHERF1 by TCF4 interaction and a promoter activity by TCF1- Nherf1 promoter gene interaction has been reported in an in vitro model of colorectal cancer [Saponaro, C et al., Oncogene 2018]. No similar studies have been conducted  on a BC model.”

TCF1 expression and Estrogen Receptor positivity

On lines 87-88 the authors state that TCF1 expression was only directly related to ER positive tumors. However, on supplementary Table 2, they report that twice as many ER-/TCF1+ than ER-/TCF1- patients.

-          We thank the reviewer for the observation. Our analysis compared the expression values of ER and TCF1 (that are reported in the column, 68 vs 42) in  the total   evaluable cases for both markers. Taking into consideration the expression of the markers present on the horizontal line (20 vs 42),  it would indeed be more appropriate to report a correlation of TCF1 with ER expression. So in the results section the sentence has been modified as follows:

TCF1 expression was directly related to ER expression (p=0.0432).

Figure 1.

Panel A – Pictures have very low resolution. Figure letters are a), c) and e) are not very clearly visible. Scale bar missing.

Panel B - Pictures have, again, low resolution and do not look sharp enough, making it very hard to appreciate cellular morphology and subcellular localization of the IF-labelled proteins. The picture legends are also small, smudged and hard to read. Scale bars seem to be present at the right-hand lower corner of each picture, but they are impossible to read.

-          We thank the reviewer for the observation.

The resolution of the pictures has been penalized because of the template format. However, the resolution and the graphical legend of the images both have been improved. The scale bars have  been placed on a white background. Further, the value of the scale bar has been added in the caption of Figure1.

cNHERF1 and nNHERF1 are only defined in line 279 and 280, yet these abbreviations are extensively used throughout the manuscript with no prior definition.

- The abbreviations have been included in  lines 79-80 of the corrected  text, as reported:

2.1. Profiling of Expression and Relationship between Tumor Markers and Clinicopathological Features

“Forty-eight per cent (106/220) of tumors presented a higher NHERF1 expression in cytoplasmic compartment (cNHERF1) as compared the 10% present in nuclear staining (nNHERF1)10% (23/220).”

Patient selection

There is no information whatsoever concerning the selection criteria, if there was any, regarding treatment regimens and therapeutic agents’ patients were submitted to.

-          In Material and Methods section we reported as follows:

Patients were eligible if they had histological diagnosis of invasive breast carcinomas of any size and no evidence of metastatic disease. Patients were excluded if they had a previous history of invasive breast cancer, or other previous or concomitant malignancies or concomitant diseases.”

All patients of our cohort were homogenously treated with chemotherapy, and the patients who were hormone status positive were treated with both chemotherapy and hormonetherapy.

Subtype-specific prognostic value of the different Wnt signalling markers

This reviewer would appreciate a BC subtype specific survival analysis for the expression of each of the Wnt markers.

- In this study we have considered the expression of different markers of the Wnt pathway, some of which are not used on tumor tissues such as breast cancer tissues. Our aim was to study the prognostic significance of these markers on the whole cohort, therefore we did not consider their prognostic value in specific subtypes. To evaluate the prognostic significance in different subgroups of the BC, we plan to expand the series in order to have a strong statistical significance.

Immunohistochemical assessment

This reviewer would like a clarification on the criteria for the selection of Pos vs Neg thresholds for the IHC of the markers described in lines 274-291.

- The median value has been used as cut-off for NHERF1 and β-catenin as has been reported in our previous works [Mangia, A et al., Oncotarget 2017; Schirosi, L et al., Oncotarget 2016]. For the other markers, because of  the absence of literature references, we  used the median value to harmonize the evaluation methods. The text has been modified as follows:

“Immunoreactivity was assessed independently by two observers, who were blinded to clinicopathological data. The results given by the two observers were identical in most cases, and discrepancies were resolved by re-examination and consensus. For all markers, NHERF1, β-catenin, FZD4, LRP5, LRP6 and TCF1 a three-field average percentage was assessed and the median values of protein expression were considered as cut-off.

NHERF1 and β-catenin immunostaining was assessed, as previously described [12,15]. NHERF1 immunostaining was predominantly cytoplasmic and in some cases an intense nuclear staining was also observed. This was evaluated and scored separately. The cases were classified as positive when cNHERF1 immunoreactivity was present in ≥80% (median value) of tumor cells, and when nNHERF1 immunoreactivity was present in >0% (median value) of tumor cells observed.

Only cytoplasmic β-catenin immunostaining was considered and the cases were classified as positive when β-catenin immunoreactivity was present in ≥ 5%  (median value) of tumor cells, nuclear staining was completely absent. FZD4 immunostaining was largely cytoplasmic. FZD4 was assessed by counting the number of immunoreactive cancer cells over total cancer cells (%) for fields (more of 500 tumor cells) at x400 magnification [34]. LRP5 and LRP6 immunoreactivity was observable as a brown cytoplasmic coloration, and a percentage of positive cells per field was reported. TCF1 was positive when strong nuclear staining was present (percentage of positive nuclei for field). For FZD4, LRP5, LRP6 and TCF1 proteins, the cases were classified positive when the immunoreactivity was present in ≥ 77%, ≥ 83% and ≥ 74% and ≥16% (median values) of tumor cells, respectively.”

English:
The manuscript is intelligible. However, it should be reviewed by a scientific-writing expert to mitigate several syntax and grammatical issues which make the read through harder. Some examples are:

i)                 the consistent lack of articles (line 17: “(The) Wnt pathway is (…)”);

ii)                the use of informal writing style in some instances (line 41: “able to interact with a lot of (…)”);

iii)              poor syntax (line 100: “Association between Protein Expressions Analyzed”); (…)

- The manuscript has been reviewed by a scientific-writing expert to improve language  register and fluency.

Reviewer 3 Report

This study was aimed to the analysis of the expression of some Wnt pathway constituents such as β-catenin, FZD4, LRP5, LRP6 and TCF1 and their relationship with protein NHERF1. The authors also evaluated the association of the examined protein expression with clinical-pathological features and survival of breast cancer patients. It was found that β-catenin, FZD4, LRP5, LRP6 and TCF1 overexpression was associated with poor prognosis of the examined cohort of breast cancer patients and TCF1 was identified as a new potential biomarker of breast cancer.  Subscribed manuscript is capable of being published with a few major revisions mainly in result part of the manuscript.

Major points

1.     Results; Paragraph 2.1. Authors should move Table S2 as a regular Table of the manuscript. This table shows important findings and significant association of the examined expression with clinical data of breast cancer patients. Furthermore, the authors could shorten description of found associations in the text.

2.     Results and Methods; There is not specified functional and molecular classification of the examined cohort of breast cancer patients. Authors should describe and divide used samples as Luminal A, Luminal B, Her2 enriched types and, if present, also Basal Like or Claudin Low types of breast carcinomas. Authors should analyse associations of the examined protein expression with those groups of breast cancer. Those findings could be very important for stratification of breast cancer and may reveal some interesting data with regard to the examined expression of proteins and their importance as biomarkers of breast cancer.

3.     Discussion; Authors should mention the biological function of the examined proteins in cells and cell pathways due to their potential importance in breast cancer.

Minor points

Abbreviation of Progesterone      Receptor: more      typical than PgR is using of PR.      It is highly recommended to use the abbreviation PR.

Author Response

All changes to our manuscript within the document have been marked in yellow.

Reviewer 3

Open Review

English language and style

( ) Extensive editing of English language and style required
( ) Moderate English changes required
(x) English language and style are fine/minor spell check required
( ) I don't feel qualified to judge about the English language and style

Yes

Can be   improved

Must be   improved

Not   applicable

Does the introduction provide sufficient background and include all   relevant references?

(x)

( )

( )

( )

Is the research design appropriate?

( )

(x)

( )

( )

Are the methods adequately described?

(x)

( )

( )

( )

Are the results clearly presented?

( )

(x)

( )

( )

Are the conclusions supported by the results?

( )

(x)

( )

( )

Comments and Suggestions for Authors

This study was aimed to the analysis of the expression of some Wnt pathway constituents such as β-catenin, FZD4, LRP5, LRP6 and TCF1 and their relationship with protein NHERF1. The authors also evaluated the association of the examined protein expression with clinical-pathological features and survival of breast cancer patients. It was found that β-catenin, FZD4, LRP5, LRP6 and TCF1 overexpression was associated with poor prognosis of the examined cohort of breast cancer patients and TCF1 was identified as a new potential biomarker of breast cancer.  Subscribed manuscript is capable of being published with a few major revisions mainly in result part of the manuscript.

Major points

1.     Results; Paragraph 2.1. Authors should move Table S2 as a regular Table of the manuscript. This table shows important findings and significant association of the examined expression with clinical data of breast cancer patients. Furthermore, the authors could shorten description of found associations in the text.

-  We have changed   Table S2,  labeling it  Table 1 as suggested by reviewer. The description of the association found has   been reported in the Results section.

“The relationship between NHERF1, β-catenin, FZD4, LRP5, LRP6, TCF1 expression and the clinicopathological characteristics are listed in Table 1. We found an increase of cNHERF1expression in tumors >2 cm (p=0.0020), and a low  nNHERF1 expression in tumors with a higher histological grade (G3), (p=0.0386). Cytoplasmic β-catenin expression was observed in IDCs (p=0.0138), and it was related to ER negative (p<0.0001), PR negative (p=0.0050), HER2/neu negative (p=0.0002) and high proliferative activity (Ki67 index) (p=0.0082). FZD4 was over-expressed in IDCs (p=0.0199), in tumors >2 cm (p=0.0010), and in tumors with a high histological grade (p=0.0004).  LRP5 was also over-expressed in tumors >2 cm (p=0.0009), in high histological grade (p=0.0002) and high Ki67 index (p=0.0049).  LRP6 was over-expressed in older patients (p=0.0175), in tumors >2 cm (p=0.0089), in G3 tumors ( p=0.0010) and high Ki67 index (p=0.0235). LRP6 was inversely related to ER (p=0.0012) and PR status (p=0.0043).  TCF1 expression was directly related to ER expression (p=0.0432) (Table 1).”

2.     Results and Methods; There is not specified functional and molecular classification of the examined cohort of breast cancer patients. Authors should describe and divide used samples as Luminal A, Luminal B, Her2 enriched types and, if present, also Basal Like or Claudin Low types of breast carcinomas. Authors should analyse associations of the examined protein expression with those groups of breast cancer. Those findings could be very important for stratification of breast cancer and may reveal some interesting data with regard to the examined expression of proteins and their importance as biomarkers of breast cancer.

- We thank the reviewer for the suggestion. 

The aim of this study was to evaluate the expression of some of major Wnt-pathway markers (beta- catenin, FZD4, LRP5, LRP6 and TCF1) in order to detect a potential association with NHERF1 protein and their clinical impact in Breast Cancer patients. Since our clinical cohort is not numerous, we did not considered the Luminal A, Luminal B, Her2 enriched subtypes etc. It would not have been possible to support the clinical /biological result  with  statistical power. Based on our results, in the future we are considering expanding  our clinical cohort to evaluate the possible association of the expression of the above markers in the subgroups suggested by the reviewer as well.

3.     Discussion; Authors should mention the biological function of the examined proteins in cells and cell pathways due to their potential importance in breast cancer.

- As suggested by reviewer, a more in-depth  discussion regarding the biological function and potential importance of examined proteins  in breast cancer has been added as follows:

“NHERF1 is an important regulator of different proteins trafficking and localization as a CF transmembrane conductance regulator, ezrin and β-catenin. It has an oncogenic role when it is localized in the plasmatic membrane, and it acts as an oncosuppressor when it moves into the cytoplasmic/nuclear compartment [Georgescu MM et al. Current Molecular Medicine, 2008; Mangia A et al., Histopathology 2009]. 

β-catenin has pivotal roles in morphogenesis and human cancer, because it has a double function both  in adhesive complexes and as a transducer/transcriptional regulator in several signal transduction pathways.  β-catenin mutations have not been found in BC, despite its upregulation and association with other markers and with the poor clinical outcome [Cowin P et al., Curr Opin Cell Biol. 2005). β-catenin signaling is initiated through the formation of a FZD/LRP5/6 receptor complex and the activation of this protein complex seems to  have a crucial role in BC development and progression.

Frizzled proteins are found at the level of the plasmatic membrane. They are sited at the surface of Wnt-responsive cells, but they have been observed co-opted in the cytoplasmic compartment as part of a regulator mechanism of the extracellular level of Wnt protein [Dubois L et al., Cell. 2001].”

Minor points

Abbreviation of Progesterone      Receptor: more      typical than PgR is using of PR.      It is highly recommended to use the abbreviation PR.

- Regarding this advice, the  abbreviation regarding “Progesterone Receptor” has been uniformly replaced by PR throughout the entire  article as well as  in all Tables.

Round 2

Reviewer 2 Report

The authors have addressed, to a certain extent, all the points of concern raised by this reviewer. This reviewer is of the opinion that the work should be accepted for publication.

Reviewer 3 Report

I have no other comments.